# The Synchronization Behaviors of Coupled Fractional-Order Neuronal Networks under Electromagnetic Radiation

**Xin Yang [1], Guangjun Zhang [2,*], Xueren Li [1] and Dong Wang [1]**

[1] Aeronautics Engineering College, Air Force Engineering University, Xi'an 710038, China; yangxin@hkgcxy.cn (X.Y.); lixueren@hkgcxy.cn (X.L.); wangdong@hkgcxy.cn (D.W.)

[2] Department of Basic Sciences, Air Force Engineering University, Xi'an 710058, China

* Correspondence: zhanggj@hkgcxy.cn

**Abstract:** Previous studies on the synchronization behaviors of neuronal networks were constructed by integer-order neuronal models. In contrast, this paper proposes that the above topics of symmetrical neuronal networks are constructed by fractional-order Hindmarsh–Rose (HR) models under electromagnetic radiation. They are then investigated numerically. From the research results, several novel phenomena and conclusions can be drawn. First, for the two symmetrical coupled neuronal models, the synchronization degree is influenced by the fractional-order $q$ and the feedback gain parameter $k_1$. In addition, the fractional-order or the parameter $k_1$ can induce the synchronization transitions of bursting synchronization, perfect synchronization and phase synchronization. For perfect synchronization, the synchronization transitions of chaotic synchronization and periodic synchronization induced by $q$ or parameter $k_1$ are also observed. In particular, when the fractional-order is small, such as 0.6, the synchronization transitions are more complex. Then, for a symmetrical ring neuronal network under electromagnetic radiation, with the change in the memory-conductance parameter $\beta$ of the electromagnetic radiation, $k_1$ and $q$, compared with the fractional-order HR model's ring neuronal network without electromagnetic radiation, the synchronization behaviors are more complex. According to the simulation results, the influence of $k_1$ and $q$ can be summarized into three cases: $\beta > 0.02$, $-0.06 < \beta < 0.02$ and $\beta < -0.06$. The influence rules and some interesting phenomena are investigated.

**Keywords:** fractional-order neuronal model; synchronization transition; neuronal network; electromagnetic radiation





## 1. Introduction

The firing behavior of neurons is a nonlinear process, and the neurons are a complex, nonlinear dynamic system. In 1952, Hodgkin and Huxley used equivalent circuits and large amounts of data from experiments to model and analyze the data, and then they constructed the Hodgkin–Huxley (HH) neuron model through theoretical derivation [1]. Then, FItzHugh, Morris and Lecar, Hindmarsh and Rose proposed the FItzHugh–Nagumo (FHN) model [2], Morris–Lecar (ML) model [3], and Hindmarsh–Rose (HR) model [4], respectively. Synchronization is an important phenomenon in the neuronal system and is one of the operational mechanisms of the brain. A number of researchers have used coupled neuronal models to try to explain some of the synchronization phenomena observed in experiments. Because the synchronization is related to neurological diseases in the brain, such as Parkinson's disease [5] and epilepsy [6], investigating the synchronization behaviors of neuronal systems by theoretical methods or experiments is helpful to understand the mechanisms of related phenomena.

### 1.1. Literature Review

For the synchronization behaviors of two coupled neuronal models and neuronal networks, many studies have been performed by relevant scholars [7–28]. For the syn-

chronization behaviors of two coupled neurons, the synchronization transition [7,8] and the influence of different coupling methods [9–11] are investigated. The synchronization transition of two coupled ML neurons was studied in [7], and bursting synchronization occurred before phase synchronization. The effects of memristive synapse coupling were investigated in [9]. When multiple neurons are coupled into a larger network, there are more emergence phenomena. Many researchers have investigated the factors influencing network synchronization [12–24], such as the network topology [12–15], the time delay and partial time delay [16,18], the coupled methods and initial value [18–21], and the layers of networks [22]. In previous studies, most neuron models were integer-order models. In [29], after analyzing the dynamics of the firing rate with a range of stimulus dynamics, the results showed that the multiple time scale adaptation is consistent with fractional-order differentiation. Fractional-order can give a more complete picture of nature than integer-order differentiation. In the past, scholars have performed much research on fraction-order dynamical systems and applied them in many fields, such as financial systems [30], biomedical systems [31,32], and the spread of infectious diseases [33], and related studied have shown the advantages of fractional-order models. The dynamic characteristics of the fractional-order Hindmarsh–Rose (HR) neuronal model were investigated in [25], and it was found that different fractional orders can induce different dynamic behaviors. For coupled fractional-order neuronal models and neuronal networks, there are few studies. For fractional-order coupled neuronal models, the two neurons reach perfect synchronization through the design of the controller in most of the literature [26–28]. In [34], it was found that a change in the fractional order can change the synchronization mode. However, the synchronization behaviors of fractional-order neuronal networks have not been investigated.

Recently, the dynamic behaviors of neurons under electromagnetic radiation have received wide attention from scholars [27,35–37]. In fact, neurons are usually exposed to electromagnetic radiation. From the experimental works [38–42], the effects of electromagnetic radiation on the neurons are being understood more clearly. Refs. [43,44] found that the distinct spike-frequency adaption will happen when the neuron is modulated by extracellular electric fields. In [36,37], the external stimulus current and external electromagnetic radiation were omitted, respectively, and the corresponding models were proposed. According to [36], the single fractional-order neuron model under electromagnetic radiation was proposed, and the dynamic behaviors were investigated in [27]. The effects of electromagnetic radiation on the dynamic characteristics and synchronization behaviors of coupled integer-order neuronal networks were investigated in [37,45–47]. The synchronization behavior of one main network coupled with some subnetworks was investigated in [45]. In [46], it was found that rhythm synchronization happens under appropriate coupling strength and electromagnetic radiation. In [27,28], the two coupled neurons with and without time-delays were in perfect synchronization by designing an appropriate controller, but the authors did not investigate the effects of parameters and the synchronization transitions.

From the above analysis, the synchronization behaviors and synchronization transitions of coupled fractional-order neuronal models and neuronal networks constructed by fractional-order HR neuronal models under electromagnetic radiation have not been investigated in previous studies. Compared with the integer-order model under electromagnetic radiation and the fractional-order models without electromagnetic radiation, the more diverse synchronization behaviors may be induced by the change in fractional-order and electromagnetic radiation. To observe more synchronization behaviors and synchronization transition modes, this paper investigated the above problems. In the studies mentioned above, the predict-corrector method [48] is used to study the fractional-order systems. The predict-corrector method has a high accuracy, but the calculation demand is large. The Adomian decomposition method (ADM) [49] used in this paper has a higher accuracy, and the calculation amount is smaller than that of the predict-corrector method [50].

### 1.2. Description of Each Section

From this perspective, this paper investigates the synchronization transition of coupled fractional-order neuronal networks under electromagnetic radiation. In Section 2, the models of coupled fractional-order neuronal networks are proposed. In Section 3, differently to [24–26], the synchronization transitions of two coupled neuronal models under electromagnetic radiation induced by the fractional-order, coupling strength and parameter of electromagnetic radiation are studied. In Section 4, to determine the influence of electromagnetic radiation, a ring network constructed by fractional-order HR models without electromagnetic radiation is constructed firstly, and the synchronization transition induced by fractional-orders is investigated. Additionally, then, this section investigates the synchronization behaviors of the fractional-order ring neuronal network constructed by fractional-order HR models under electromagnetic radiation under three conditions.

## 2. Model Description

There are three definitions which are the most frequently used: the Grunwald–Letnikov, Riemann–Liouville and Caputo derivatives. The Caputo is simpler to solve the fractional-order derivative. This paper adopted the Caputo derivative to investigate the fractional-order systems.

**Definition 1.** *The Caputo derivative of the function* $f(x)$ *is defined as*

$$
{}_0^C D_t^q f(x) \; = \; \frac{1}{\Gamma(n-q)} \int_0^t \frac{f^{(n)}(\tau)}{(t-\tau)^{q-n+1}} d\tau
$$

*where* $n-1 < q < n$ *and* $\Gamma(\bullet)$ *is the gamma function, which is defined as*

$$
\Gamma(z) \; = \; \int_0^\infty t^{z-1} e^{-1} dt
$$

*In particular, when* $0 < q < 1$,

$$
{}_0^C D_t^q f(x) \; = \; \frac{1}{\Gamma(1-q)} \int_0^t \frac{f\prime(\tau)}{(t-\tau)^q} d\tau
$$

In this paper, to analyze the dynamic behaviors of the fractional-order neuronal network, the HR model is adopted for a single neuronal model. The HR model is described as follows [25]:

$$
\begin{cases}
D_t^q x \;=\; y - ax^3 + bx^2 - z + I_{exc} \\
D_t^q y \;=\; c - dx^2 - y \\
D_t^q z \;=\; r[s(x - \overline{x}) - z]
\end{cases}
\tag{1}
$$

where $x$ is the membrane action potential, $y$ is a recovery variable, $z$ is a slow adaption current, $D_t^q$ is the differential operator defined by Caputo, and $q$ is the fractional-order. $I_{exc}$ is the external stimulus current. In this paper, parameters except for the fractional-order and external stimulus current are fixed as $a = 1$, $b = 3$, $c = 1$, $d = 5$, $r = 0.006$, $\overline{x} = -1.56$, $s = 4$ [32].

The two coupled fractional-order neuronal models under electromagnetic radiation can be described as follows [35]:

$$
\begin{cases}
D_t^q x_{1,2} \;=\; y_{1,2} - ax_{1,2}{}^3 + bx_{1,2}{}^2 - z_{1,2} + I_{exc} + k_1 W(\varphi_{1,2}) x_{1,2} + C(x_{2,1} - x_{1,2}) \\
D_t^q y_{1,2} \;=\; c - dx_{1,2}{}^2 - y_{1,2} \\
D_t^q z_{1,2} \;=\; r[s(x_{1,2} - \overline{x}) - z_{1,2}] \\
D_t^q \varphi_{1,2} \;=\; x_{1,2} - k_2 \varphi_{1,2} + \varphi_0
\end{cases}
\tag{2}
$$

$\varphi$ denotes the magnetic flux across the cell membrane. $\varphi_0$ is the external magnetic flux. $k_1$ is the feedback gain. The cubic flux-controlled memristor model $W(\varphi) = dq(\varphi)/d\varphi = \alpha + 3\beta\varphi^2$ is introduced in this model, and it is used to estimate the effect of feedback regulation on

membrane potential when the magnetic flux is changed. The two parameters, $\alpha$ and $\beta$, describe the memory conductance, and they vary with the environment and their own conditions. In this paper, $\alpha = 0.2, k_2 = 0.4, \varphi_0 = 1$.

The fractional-order neuronal ring network without electromagnetic radiation can be described as follows:

$$\begin{cases} D^q x_i = y_i - ax_i{}^3 + bx_i{}^2 - z_i + I_{exc} + \frac{C}{2P} \sum_{j=i-1}^{i+1} a_{ij}(x_j - x_i) \\ D^q y_i = c - dx_i{}^2 - y_i \\ D^q z_i = r[s(x_i - \overline{x}) - z_i] \end{cases} \tag{3}$$

where $i = 1, 2, \ldots, N$ and $C$ is the coupling strength of the network. If node $i$ and node $j$ are connected, $a_{ij} = a_{ji} = 1$. Each neuronal model is symmetrically coupled to its 2P nearest neighbors. It is important to note that $j = -m, (m = 1, 2, \ldots)$ implies that node $j$ is coupled with node $(101 - m)$. The network structure is shown in Figure 1.

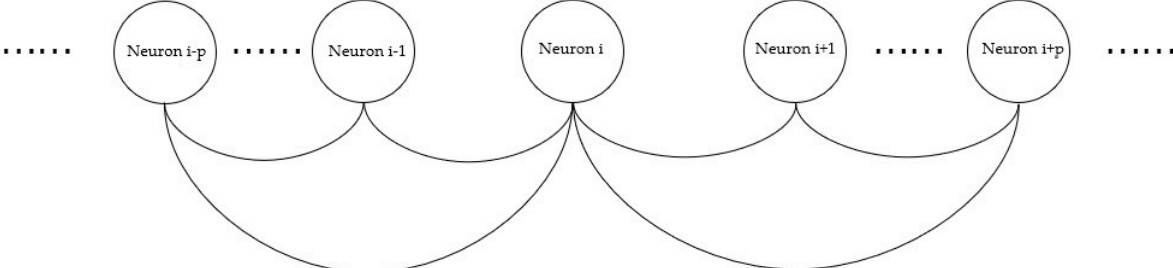

**Figure 1.** Structure of the neuronal network.

Then, the fractional-order neuronal ring network under electromagnetic radiation can be described as follows:

$$\begin{cases} D_t^q x_i = y_i - ax_i{}^3 + bx_i{}^2 - z_i + I_{exc} + k_1 W(\varphi_i)x_1 + \frac{C}{2P} \sum_{j=i-1}^{i+1} a_{ij}(x_j - x_i) \\ D_t^q y_i = c - dx_i{}^2 - y_i \\ D_t^q z_i = r[s(x_i - \overline{x}) - z_i] \\ D_t^q \varphi_i = x_i - k_2 \varphi_i + \varphi_0 \end{cases} \tag{4}$$

## 3. Synchronization Behavior of Two Fractional-Order Coupled Neuronal Models under Electromagnetic Radiation

As the simplest neuronal network, the synchronization behaviors of two coupled fractional-order neuronal models under electromagnetic radiation are researched first.

To prove that the two coupled fractional-order neuronal models can come into perfect synchronization, some lemmas and definitions are introduced as follows. The Mittag–Leffler function is defined by

$$E_{\alpha,\beta}(z) := \sum_{i=0}^{\infty} \frac{z^i}{\Gamma(\alpha i + \beta)}$$

**Lemma 1.** [51] *Let $x(t) \in C^m$ be a real continuous and differentiable vector function. Then, for all $t \geq t_0$ and $0 < q < 1$, the following inequality holds:*

$$D^q\left(x^H(t)Px(t)\right) \leq x^H(t)P(D^q x(t)) + \left(D^q x^H(t)\right)Px(t)$$

**Lemma 2.** [52] *Let $V(t)$ be a continuous function on $[t_0, +\infty)$ that satisfies*

$$D^q V(t) \leq \theta V(t)$$

*where $0 < q < 1$ and $\theta$ are constants, then*

$$V(t) \leq V(t_0) E_\alpha \left( \theta (t - t_0)^\alpha \right)$$

**Lemma 3.** [53] *For $0 < q < 1$, $t \in \mathbb{R}$, $t > 0$, we have*

$$\lim_{t \to \infty} E_q(t) \leq \lim_{t \to \infty} \frac{1}{q} e^{t^{\frac{1}{q}}}$$

Let $X = (x_1, y_1, z_1, \varphi_1)$, $Y = (x_2, y_2, z_2, \varphi_2)$, $C = (C, 0, 0, 0)$. *The error system is $e = X - Y$. According to (2), the error system can be described as*

$$e = f(X) - f(Y) - 2C(X - Y)$$

*where*

$$f(X) = \begin{bmatrix} y_1 - ax_1^3 + bx_1^2 - z_1 + I_{exc} + k_1 W(\varphi_1)x_1 \\ c - dx_1^2 - y_1 \\ r[s(x_1 - \overline{x}) - z_1] \\ x_1 - k_2\varphi_1 + \varphi_0 \end{bmatrix}$$

$$f(Y) = \begin{bmatrix} y_2 - ax_2^3 + bx_2^2 - z_2 + I_{exc} + k_1 W(\varphi_2)x_2 \\ c - dx_2^2 - y_2 \\ r[s(x_2 - \overline{x}) - z_2] \\ x_2 - k_2\varphi_2 + \varphi_0 \end{bmatrix}$$

**Theorem 1.** $f(\bullet)$ *satisfyiesthe Lipschitz condition $\|f(X) - f(Y)\| \leq L\|X - Y\|$ [54] if*

$$L - 2C < 0$$

*Then, the two neuronal models can undergo global exponential synchronization.*

**Proof of Theorem 1.** Construct the Lyapunov function as $V = e^T e$. From Lemma 1, the Lyapunov function can be reduced to

$$
\begin{aligned}
D^q V(t) \;&\leq\; e^T D^q e + (D^q e^T) e \\
&= e^T (f(X) - f(Y) - 2C(X - Y)) + (f(X) - f(Y) - 2C(X - Y))^T e \\
&= e^T (f(X) - f(Y)) - 2e^T C(X - Y) + (f(X) - f(Y))^T e - 2(C(X - Y))^T e \\
&\leq e^T (L\|X - Y\|) - 2Ce^T e + (L\|X - Y\|)^T e - 2Ce^T e \\
&\leq 2L\|e\|^2 - 4C\|e\|^2 \\
&= (2L - 4C)V(t)
\end{aligned}
$$

From Lemma 2, $V(t) \leq V(t_0) E_q\left((2L - 4C)(t - t_0)^q\right)$, according to Lemma 3,

$$\lim_{t \to \infty} E_q\left((2L - 4C)(t - t_0)^q\right) \leq \lim_{t \to \infty} \frac{1}{q} e^{(t - t_0)(2L - 4C)^{\frac{1}{q}}}$$

Therefore, $V(t) \leq V(t_0)\frac{1}{q} e^{(t - t_0)(2L - 4C)^{\frac{1}{q}}}$ as $t \to \infty$. When $L - 2C < 0$, we can conclude that the two neuronal modes can be in global exponential synchronization when under the appropriate coupling strength. $\square$

The above condition is just sufficient, so it is necessary to investigate the details of the system's synchronization behaviors and synchronization transition.

Fractional-order HR neuronal models under electromagnetic radiation were investigated in [27], and two neuronal models reached perfect synchronization through adaptive control. However, the synchronization behavior and synchronization transition induced by the fractional-order and other parameters are not reported in [27]. In this section, the synchronization behaviors and synchronization transitions induced by the fractional-order and the parameter $k_1$ are studied.

In this section, the similarity function is introduced to measure the synchronization of the system [7]. The model of the similarity function can be described as follows:

$$S = \left[ \frac{\left\langle (x_1(t) - x_2(t))^2 \right\rangle}{\left( \langle x_1^2(t) \rangle \langle x_2^2(t) \rangle \right)^{\frac{1}{2}}} \right]^{\frac{1}{2}} \tag{5}$$

where $\langle \bullet \rangle$ stands for the time average.

Obviously, perfect synchronization can be observed when $S$ is equal to 0, and the larger $S$ is, the worse the synchronization.

The phase synchronization is introduced, and its synchronization degree is weaker than perfect synchronization. From the time of sampled time series $(t_1, t_2, \ldots, t_n)$ across the Poincare section, the phase of the neuronal model can be calculated. The phase is calculated by [55]

$$\varphi = 2\pi \frac{t - t_i}{t_{i+1} - t_i} + 2\pi i, t_i < t < t_{i+1} \tag{6}$$

The phase difference between two neuronal models is defined by

$$\Delta \varphi = |\varphi_1 - \varphi_2|$$

when the system is not in perfect and phase synchronization, it may be in bursting synchronization. Bursting synchronization means rhythm synchronization of slow variables, so bursting synchronization can be measured by the slow variable's similarity function [55]. The model of the similarity function can be described as follows:

$$S_z = \left[ \frac{\left\langle (z_1(t) - z_2(t))^2 \right\rangle}{\left( \langle z_1^2(t) \rangle \langle z_2^2(t) \rangle \right)^{\frac{1}{2}}} \right]^{\frac{1}{2}} \tag{7}$$

when $S_z$ is close to 0, the system reaches bursting synchronization.

*3.1. Effect of Fractional-Order and Coupling Strength on the Synchronization under Electromagnetic Radiation*

When $k_1 = 0.2$, the fractional-order $q$ and parameter $C$ are varied in the regions $[0.55, 1]$ and $[0.1, 0.5]$, the similarity function is shown in Figure 2, where the blue region with S equal to 0 denotes that the system is in perfect synchronization. As shown in Figure 2, when $0.55 < q < 0.67$ and the coupling strength is approximately 0.367, the similarity function undergoes a complex change, which means that the synchronous threshold of the coupling strength changes greatly. When $0.67 < q < 1$, the synchronous threshold of the coupling strength decreases with increasing fractional order. When $0.366 < C < 0.378$, the $S \sim q$ curves are shown in Figure 3. In [34], the threshold of the coupling strength only increases first and then decreases with increasing fractional-order.

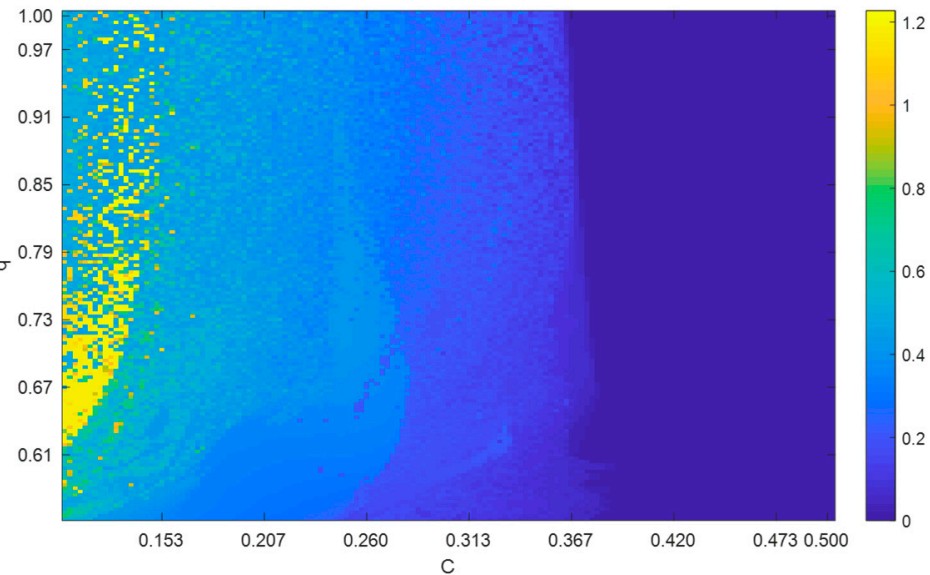

**Figure 2.** Distribution of the similarity function in the $[q, C]$ plane.

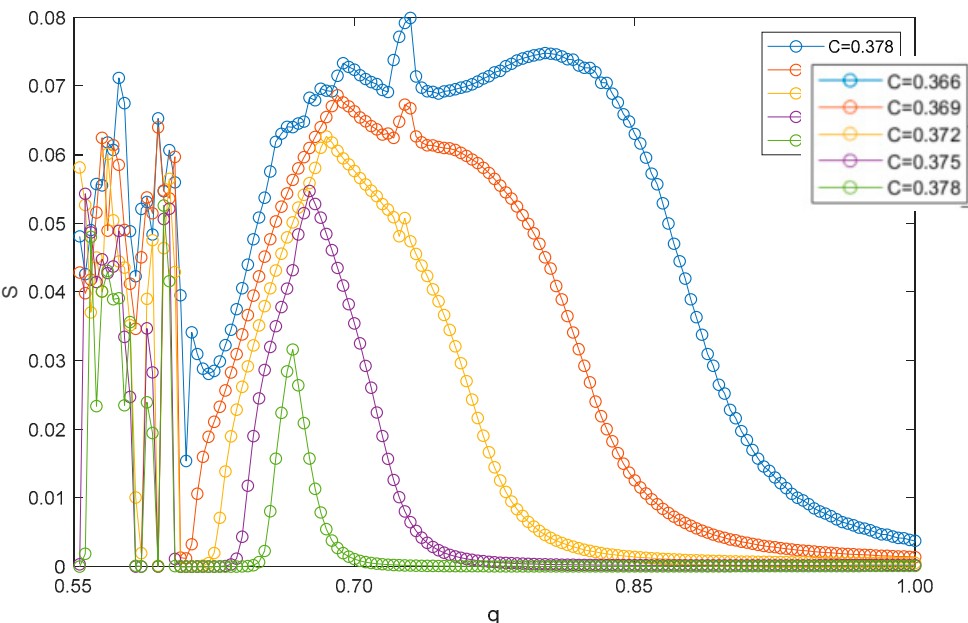

**Figure 3.** Curves of $S \sim q$ for different coupling strengths.

As shown in Figure 2, the similarity function has striking changes when $0.366 < C < 0.378$, so the synchronization transition can be observed when $0.366 < C < 0.378$. The synchronization transition when $C = 0.372$ is taken as an example to observe the change in the synchronization mode. When $q = 0.56$, Figure 4a is the phase diagram of $(x_1, x_2)$, and Figure 4b is the corresponding time series of $x_1$ and $x_2$, so the system is in imperfect synchronization. As shown in Figure 4c,f, the system is in perfect synchronization when $q = 0.61$ and $q = 0.9$ because the phase plane is located on a three quadrant angular bisector. The system is in imperfect synchronization when $q = 0.7$, as shown in Figure 4d,e, and the neuronal models display spiking.

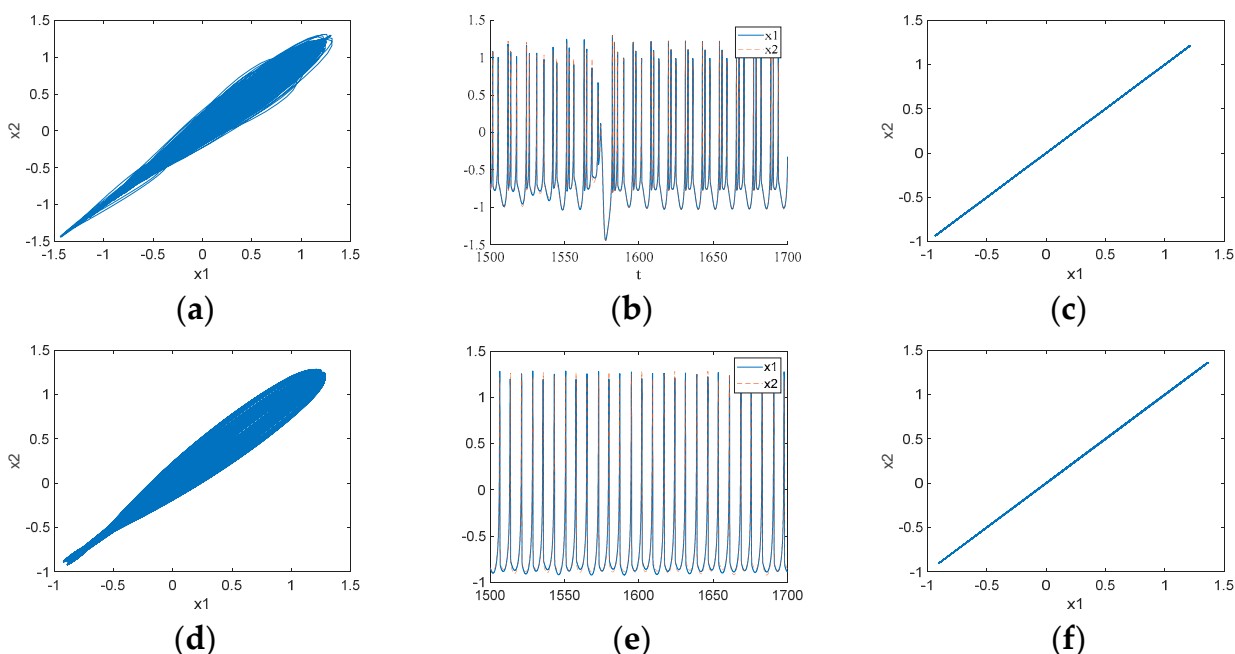

**Figure 4.** Phase diagrams of $(x_1, x_2)$ for (**a**) $q = 0.56$, (**c**) $q = 0.61$, (**d**) $q = 0.7$, and (**f**) $q = 0.9$. The corresponding time series of $x$ for (**b**) $q = 0.56$, and (**e**) $q = 0.7$.

From Figure 4, when the two neuronal models are in imperfect synchronization, the phase difference is calculated as follows. As shown in Figure 5a, when $q = 0.56$ the phase differences are always approximately 0 or $2\pi$, so the system achieves phase synchronization. From Figure 5b, the phase difference is approximately 0, so the system is also in phase synchronization when $q = 0.7$.

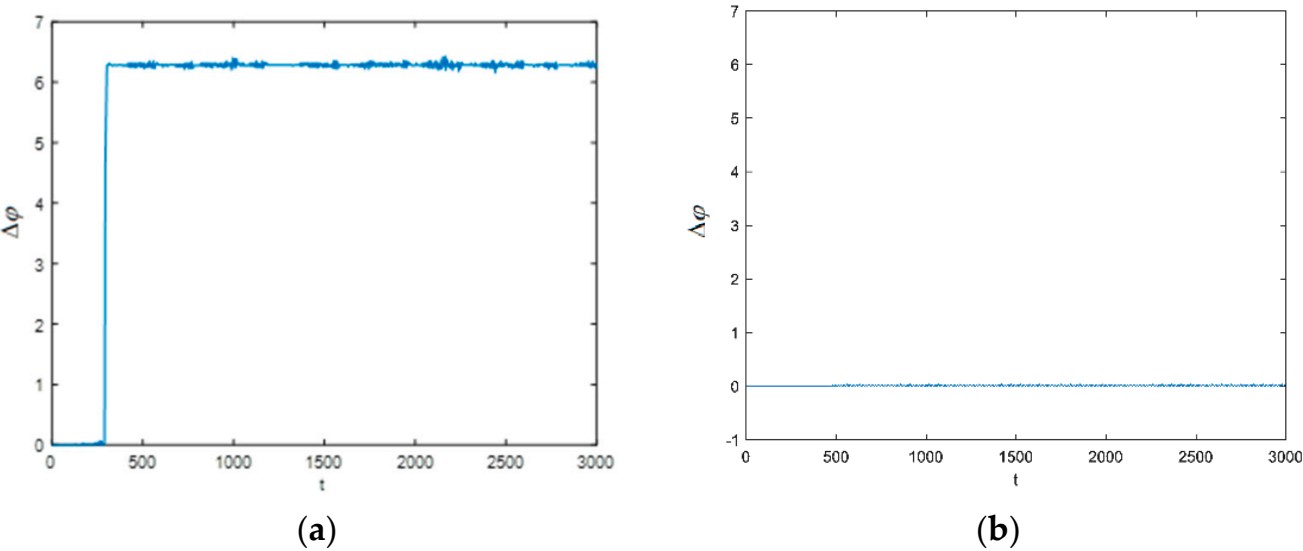

**Figure 5.** Phase difference variation with time (**a**) $q = 0.56$, (**b**) $q = 0.7$.

As mentioned above, the synchronization transition mode induced by fractional order is phase synchronization $\rightarrow$ perfect synchronization $\rightarrow$ phase synchronization $\rightarrow$ perfect synchronization. In [34], the synchronization transition near the threshold of coupling strength is perfect synchronization $\rightarrow$ phase synchronization $\rightarrow$ perfect synchronization. The transition mode in this paper is more complex than the coupled fractional-order neuronal models without electromagnetic radiation [34].

As shown in Figure 2, the two neuronal models are in perfect synchronization when $C = 0.5$ and $0.55 < q < 1$. Now, the ISI bifurcation of the first neuronal model for $C = 0.5$ is shown in Figure 6. Double-period bifurcation exists when $q = 0.66$ and $q = 0.61$ with decreasing fractional-order. The system is in chaotic perfect synchronization when $0.57 < q < 0.59$, but the system is in periodic perfect synchronization at other fractional orders. Figure 7 shows the phase diagram of $(z_1, x_1)(z_2, x_2)$ when $q = 0.56$, $q = 0.58$, $q = 0.6$, $q = 0.65$, and $q = 0.9$ for $C = 0.5$. The neuronal models display periodic-3 bursting, chaotic bursting, periodic-4 bursting, periodic-2 bursting, and spiking. The synchronization transition, which is perfect periodic-3 synchronization → perfect chaotic synchronization → perfect periodic-4 synchronization → perfect periodic-2 synchronization → perfect spiking synchronization, is observed. When the system is in perfect synchronization, the transition mode is also more complex than coupled neuronal models without electromagnetic radiation [34], because the synchronization transition is only perfect periodic-4 synchronization → perfect chaotic synchronization in [34]. The results show that when the system is in perfect synchronization, the fractional-order and electromagnetic radiation can also change the synchronization mode.

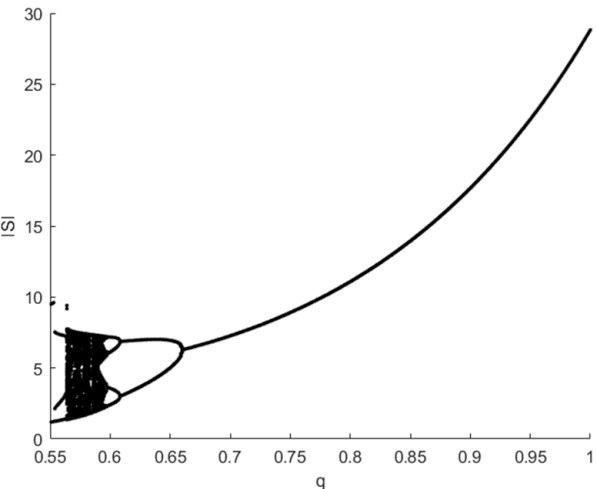

**Figure 6.** ISI bifurcation of the first neuronal model.

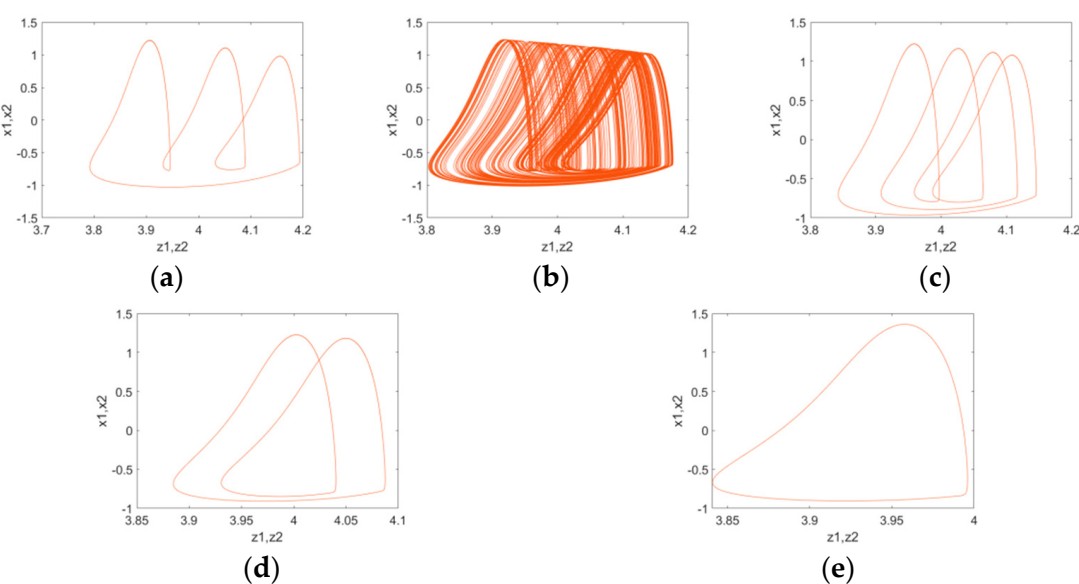

**Figure 7.** Phase diagram of $(x_1, z_1)$ and $(x_2, z_2)$ for (**a**) $q = 0.56$, (**b**) $q = 0.58$, (**c**) $q = 0.6$, (**d**) $q = 0.65$, and (**e**) $q = 0.9$.

### 3.2. Effect of the Parameter $k_1$ and the Coupling Strength on the Synchronization under Different Fractional Order

In this section, the influence of parameter $k_1$ and coupling strength on the synchronization behaviors and the synchronization transition induced by $k_1$ is investigated. The similarity function, phase difference, and slow variable similarity function are also used to measure the synchronization degree of the system. From the analysis mentioned above, different fractional orders induce different dynamic behaviors. The influence of parameter $k_1$ and coupling strength on synchronization are studied under different fractional-orders.

Therefore, the value of the similarity function in the $(C, k_1)$ plane for several fractional-order values is calculated, as shown in Figure 8. In this figure, the blue region where $S$ equals 0 denotes a system in perfect synchronization. Generally, the conclusion that a large parameter $k_1$ and coupling strength can bring the system into perfect synchronization can be drawn. However, there is an exception: when $q = 0.8$ and $q = 0.95$, a smaller $k_1$ can cause the system to be in perfect synchronization, but a larger $k_1$ cannot. As shown in Figure 9a, when $q = 0.8, C = 0.34, 0.1 < k_1 < 0.27$, the value of the similarity function decreases with increasing $k_1$, and then $S = 0$ when $0.27 < k_1 < 0.35$. However, with the sequential increase in $k_1$, $S > 0$ when $0.35 < k_1 < 0.39$. Then, $S = 0$ when $k_1 > 0.39$. To observe the above process more visually, the phase diagram and the corresponding time series are shown in Figure 9b–f when $q = 0.8, C = 0.34$. As shown in Figure 9b,c, the phase diagram of $(x_1, x_2)$ and the corresponding time series of $x_1$ and $x_2$ show that the system is in imperfect synchronization when $k_1 = 0.2$. From Figure 9d,f, the system is in perfect synchronization when $k_1 = 0.3$ and $k_1 = 0.4$. The system is asynchronized when $k_1 = 0.37$, as shown in Figure 9e.

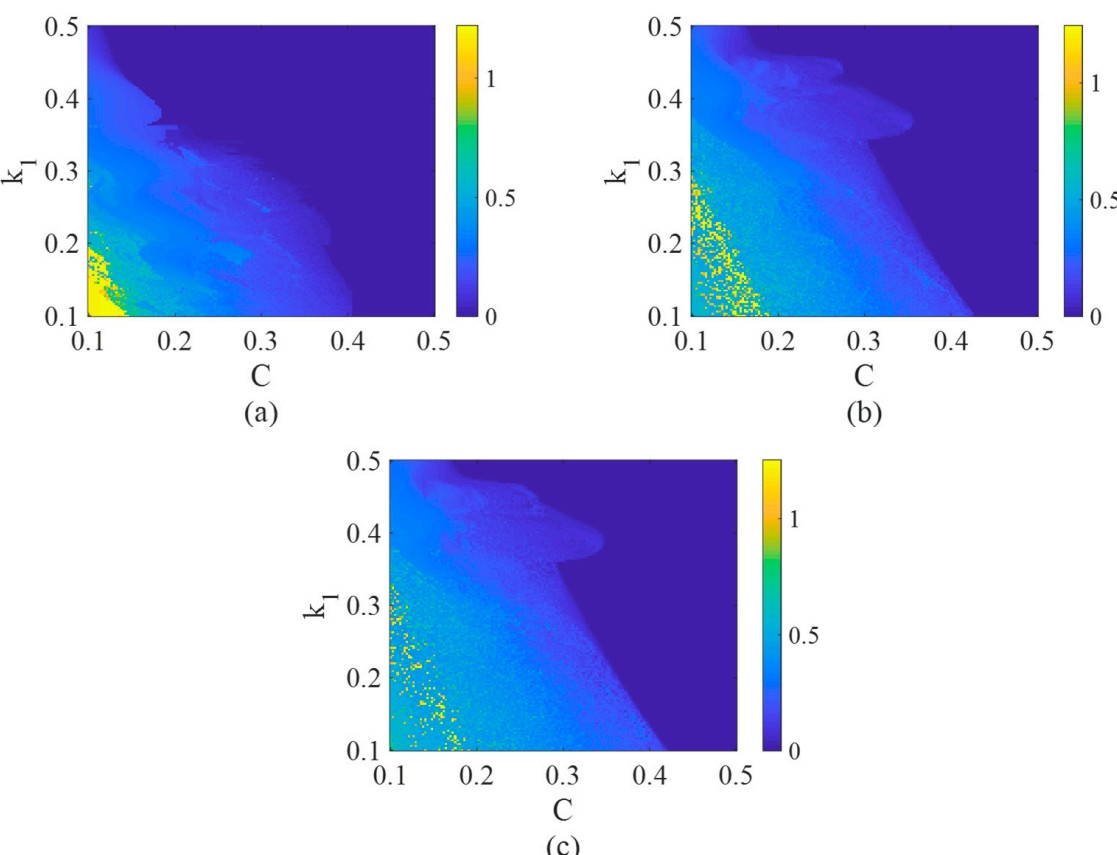

**Figure 8.** Distribution of the similarity function in the $k_1 - C$ plane for (**a**) $q = 0.6$, (**b**) $q = 0.8$, and (**c**) $q = 0.95$.

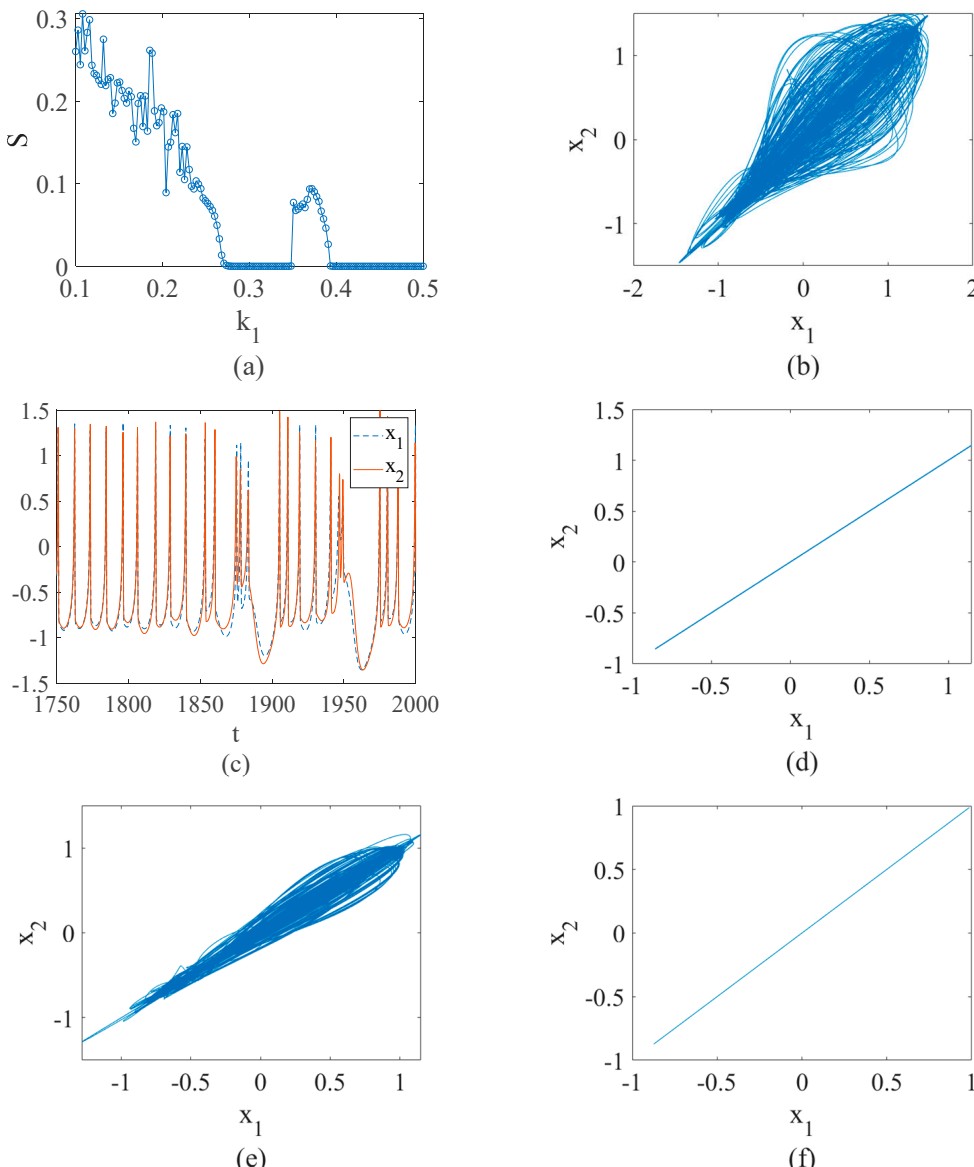

**Figure 9.** (**a**) Curve of $S \sim k_1$ for $C = 0.34, q = 0.8$. The phase diagram of $(x_1, x_2)$ for (**b**) $k_1 = 0.2$, (**d**) $k_1 = 0.3$, (**e**) $k_1 = 0.37$, and (**f**) $k_1 = 0.4$. (**c**) Corresponding time series of $x$ for $k_1 = 0.37$.

The phase differences of $x_1$ and $x_2$ when $k_1 = 0.2$ and $k_1 = 0.37$ are plotted in Figure 10. We can conclude that the two neuronal models are in phase synchronization when $k_1 = 0.37$, because the phase difference is small at approximately 0. The two neuronal models are not in phase synchronization when $k_1 = 0.2$, but the phase is locked in some time periods, and the slow variable's similarity function is calculated as $S_z = 0.05$, so the two neuronal models are in bursting synchronization. The synchronization transition induced by parameter $k_1$, that is, bursting synchronization $\rightarrow$ perfect synchronization $\rightarrow$ phase synchronization $\rightarrow$ perfect synchronization, can be observed.

Figure 8 also shows that the range of synchronization when $q = 0.6$ is the largest of the three values of $q$. As shown in Figure 11, $C = 0.26, k_1 = 0.38$, the system is asynchronized when $q = 0.8$ and $q = 0.95$, but the system is perfectly synchronized when $q = 0.6$.

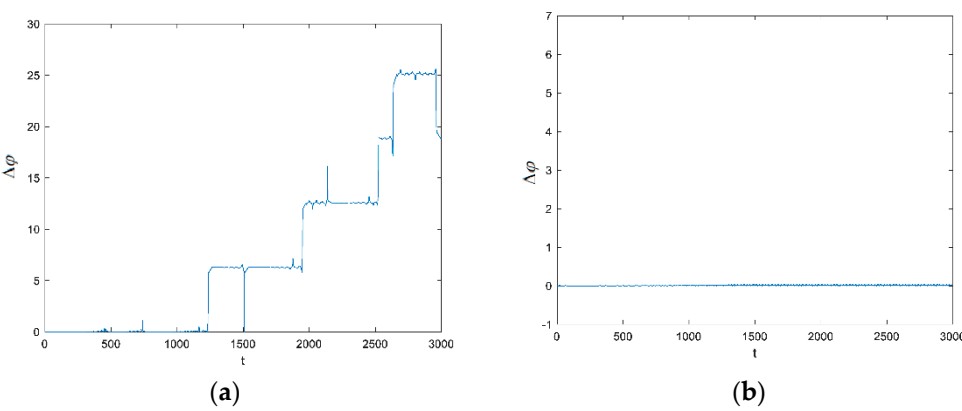

**Figure 10.** Phase difference varied with time (**a**) $k_1 = 0.2$, (**b**) $k_1 = 0.37$.

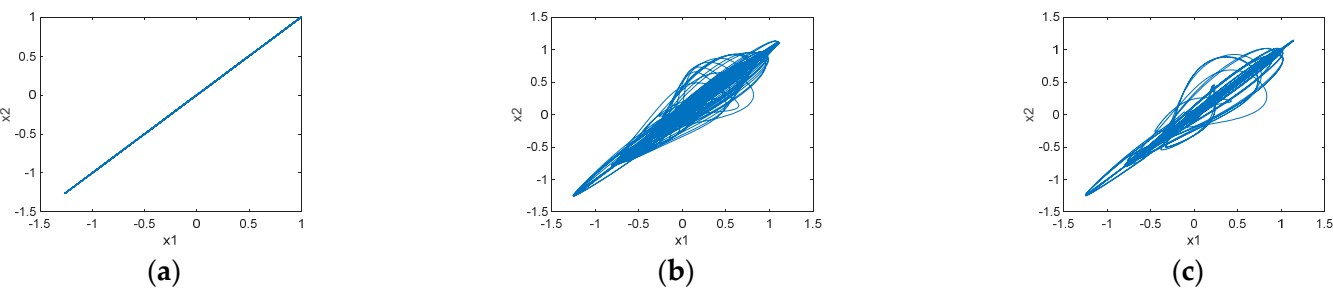

**Figure 11.** Phase diagram of $(x_1, x_2)$ for (**a**) $q = 0.6$, (**b**) $q = 0.8$, and (**c**) $q = 0.95$.

If we change the parameter $k_1$ when the system is in perfect synchronization ($C = 0.5$) for several values of $q$, the synchronization mode also changes. The ISI bifurcation of the first neuronal model is plotted in Figure 12. As shown in Figure 12a, when $0.1 < k_1 < 0.14$ and $0.25 < k_1 < 0.33$, the neuronal models display chaotic firing, but when $0.14 < k_1 < 0.24$ and $k_1 > 0.33$, the neuronal models display periodic firing. When $q = 0.6, C = 0.5$, the parameter $k_1$ induces the transition of chaotic perfect synchronization and periodic perfect synchronization. As shown in Figure 12b, when $q = 0.8, C = 0.5$, there is a double-period bifurcation when $k_1 = 0.38$. $k_1$ can also induce the synchronization transition. Thus, the other kind of synchronization transition, in which the firing modes of the two perfect synchronization neuronal models are different, induced by $k_1$ is also observed.

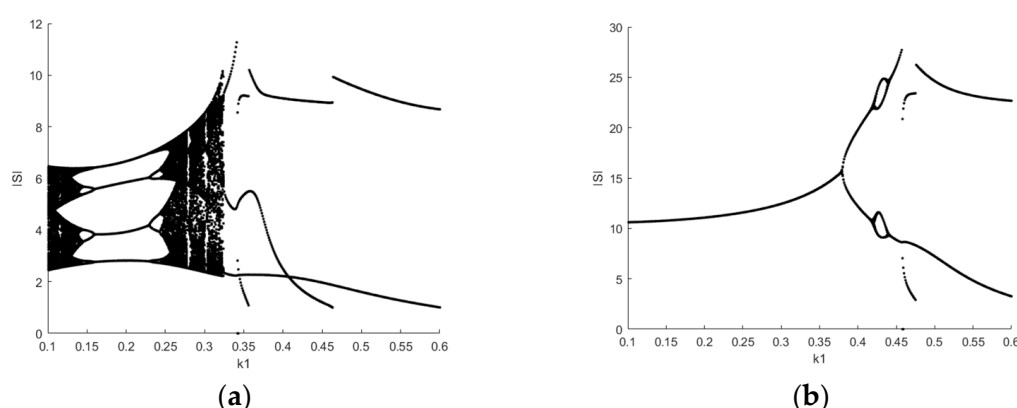

**Figure 12.** ISI bifurcation of the first neuronal model for $C = 0.5$ (**a**) $q = 0.6$, (**b**) $q = 0.8$.

The phase diagrams of $(z_1, x_1)(z_2, x_2)$ are shown in Figure 13 when $C = 0.5$. The synchronization transition induced by the parameter $k_1$ can be explained. Figure 13a,c show that the neuronal models display chaotic firing, so the system is in chaotic synchronization when $q = 0.6, k_1 = 0.12$ and $q = 0.6, k_1 = 0.3$. From Figure 13b,d, the neuronal

models display periodic-4 bursting and periodic-3 bursting, so the system is in periodic synchronization when $q = 0.6, k_1 = 0.16$ and $q = 0.6, k_1 = 0.35$. The system is in periodic synchronization at all values of $k_1$ when $q = 0.8$, but the neuronal models display spiking firing and periodic-2 bursting when $k_1 = 0.15$ and $k_1 = 0.4$, as shown in Figure 13e,f.

In [27], the effect of parameter $k_1$ on the synchronization was not investigated. In [46], when the neuronal model is an integer-order model, the synchronization mode is perfect spiking synchronization for all $k_1$ when $C = 0.5$. The conclusion can be summarized that parameter $k_1$ can induce the synchronization transition and that the synchronization behaviors are different when the fractional order varies. As shown in Figures 12 and 13, a small fractional order, such as 0.6, can induce a more complex synchronization transition because when $q = 0.6$, the system has diverse synchronization modes such as chaotic synchronization and different kinds of periodic synchronization, but only periodic synchronization occurs when $q = 0.8$.

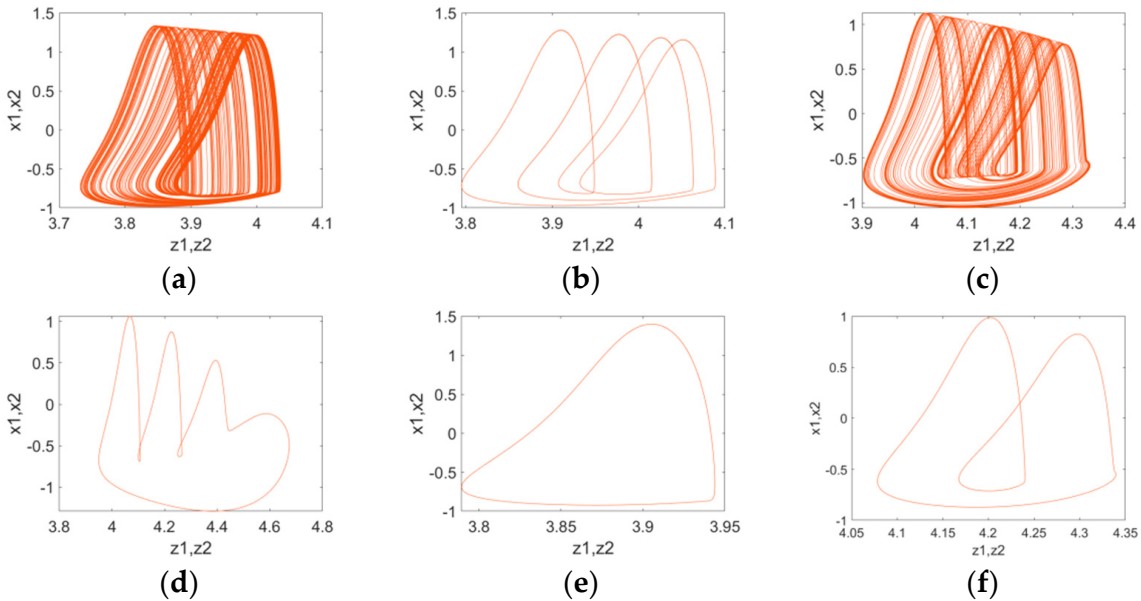

**Figure 13.** The phase diagram of $(x_1, z_1)$ and $(x_2, z_2)$ for $C = 0.5$ (**a**) $q = 0.6, k_1 = 0.12$, (**b**) $q = 0.6, k_1 = 0.16$, (**c**) $q = 0.6, k_1 = 0.3$, (**d**) $q = 0.6, k_1 = 0.35$, (**e**) $q = 0.8, k_1 = 0.15$, (**f**) $q = 0.8, k_1 = 0.4$.

## 4. Synchronization Behavior of Fractional-Order Neuronal Ring Networks under Electromagnetic Radiation

In this paper, the synchronization factor $R$ is adopted to describe the synchronization of network. $R$ is given by [47]

$$R = \frac{\langle F^2 \rangle - \langle F \rangle^2}{\frac{1}{N}\sum_{i=1}^{N}\left(\langle x_i^2 \rangle - \langle x_i \rangle^2\right)}, F = \frac{1}{N}\sum_{i=1}^{N} x_i \tag{8}$$

where $\langle \bullet \rangle$ denotes the time averaging. The value of $R$ is between 0 and 1, and it increases with decreasing average membrane potential errors. More precisely, perfect synchronization is expected when $R$ is close to 1, and a nonsynchronization state may appear when $R$ is close to 0.

The bursting synchronization of the neuronal network can be described by the slow variable's synchronization factor $R_z$. $R_z$ is given by

$$R_z = \frac{\langle F_z^2 \rangle - \langle F_z \rangle^2}{\frac{1}{N} \sum_{i=1}^{N} \left( \langle z_i^2 \rangle - \langle z_i \rangle^2 \right)}, F_z = \frac{1}{N} \sum_{i=1}^{N} z_i \qquad (9)$$

where $\langle \bullet \rangle$ denotes the time averaging. When $R_z$ is close to 1, the system reaches bursting synchronization.

### 4.1. Fractional-Order Neuronal Ring Network without Electromagnetic Radiation

To illustrate the diverse synchronization behaviors and synchronization transition modes, a fractional-order neuronal ring network without electromagnetic radiation is studied first.

The ring network consists of 100 nodes. Numerical simulation is provided for model (2) by utilizing the ADM method. As shown in Figure 14, the synchronous threshold of the coupling strength varies with the fractional-order. The change trend of the threshold first increases and then decreases, as shown in Figure 14b. In previous studies, for integer-order neuronal networks, the threshold of the coupling strength can change with other influencing factors [12,16], such as the structure of the network and time delay. In this paper, the fractional order is proven to be an influencing factor.

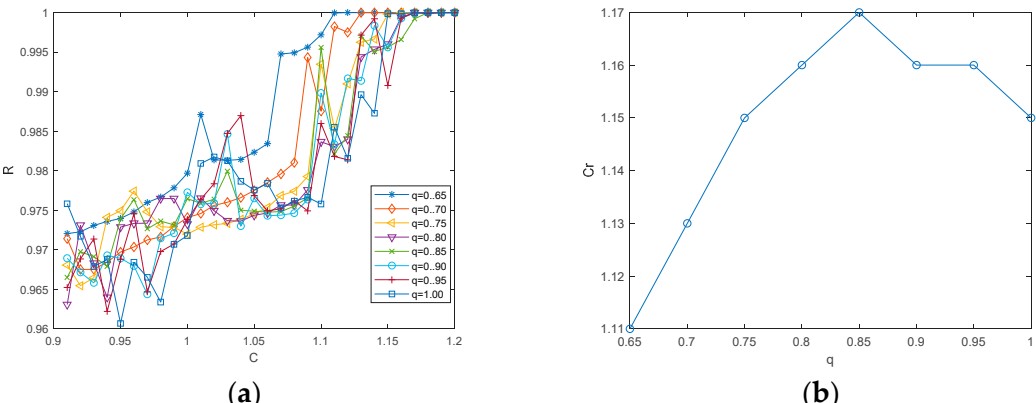

**Figure 14.** (**a**) Curves of $R \sim C$ for different fractional-orders. (**b**) The curve of $C_r \sim q$.

As shown in previous studies, for the integer-order neuronal network, a time-delay can induce the synchronization transition of the neuronal network [16,17]. In this paper, the results show that fractional-order not only makes the synchronous system become asynchronous, but it also changes the synchronization mode. When the neuronal network is in perfect synchronization, Figure 15 shows the phase diagram of $(z_i, x_i)(i = 1, 2, \ldots, N)$ when the fractional order is 0.65, 0.75, 0.85, and 0.95. The neurons display periodic-6 bursting, periodic-10 bursting, and periodic-5 bursting when $q = 0.65$, $q = 0.75$, and $q = 0.85$, as shown in Figure 15a–c, respectively. The network is in perfect periodic-6, perfect periodic-10 and perfect periodic-5 synchronization. When $q = 0.95$, as shown in Figure 15d, the neurons display chaotic bursting, so the network is in perfect chaotic synchronization. The fractional-order $q$ can induce the synchronization transition, which is perfect periodic-6 synchronization $\rightarrow$ perfect periodic-10 synchronization $\rightarrow$ perfect periodic-5 synchronization $\rightarrow$ perfect chaotic synchronization.

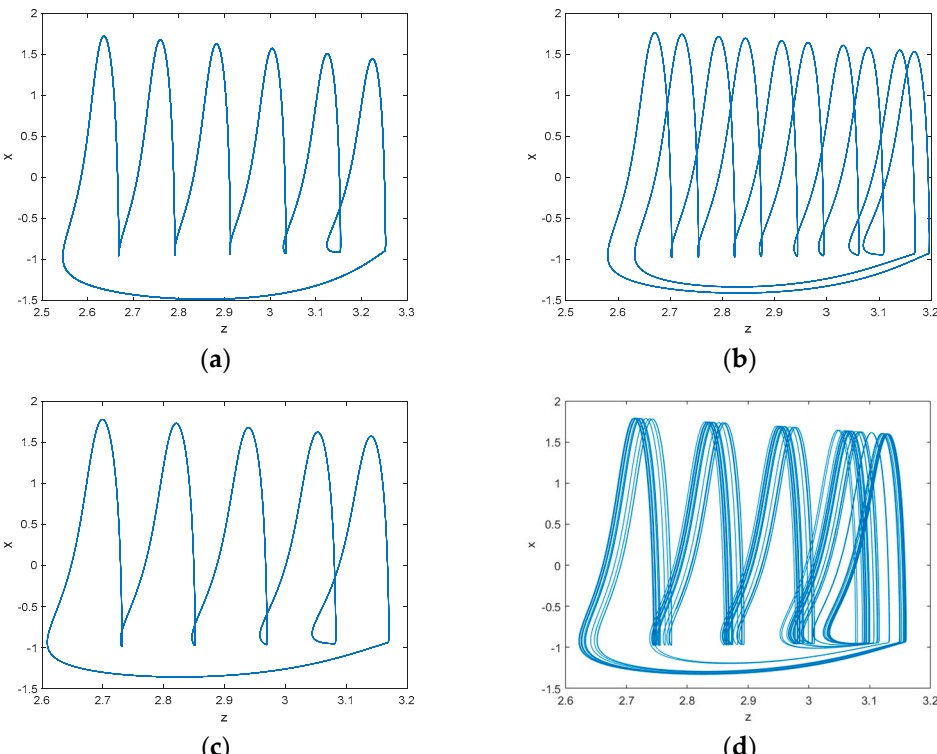

**Figure 15.** Phase diagrams of $(z_i, x_i)(i = 1, 2, \ldots, N)$ for (**a**) $q = 0.65$, (**b**) $q = 0.75$, (**c**) $q = 0.85$, and (**d**) $q = 0.95$.

### 4.2. Fractional-Order Neuronal Network under Electromagnetic Radiation

In this section, the effect of some parameters on the synchronization behaviors of ring neuronal networks under electromagnetic radiation is investigated. In [37], the integer-order neuron's firing activity is influenced by the parameter $k_1$. In this paper, the firing activity is also influenced by the parameter $\beta$ apart from $k_1$. In addition, in this paper, the results show that the fractional order cannot influence the neuronal models' firing mode in the ring neuronal network under electromagnetic radiation. It is obviously different from the two coupled fractional-order neuronal models under electromagnetic radiation, single neuron and the ring neuronal network without electromagnetic radiation. Although the fractional-order cannot influence the neuronal firing mode, the fractional-order can influence the synchronization degree of the ring neuronal network under electromagnetic radiation under different conditions. For example, the ring neuronal network constructed by integer-order neuronal models is in perfect synchronization, but the perfect synchronization may be destroyed in the same conditions when the integer-order neuronal models are instead fractional-order neuronal models, or vice versa. Therefore, this paper focuses on the effects of parameters $\beta$, $k_1$ and $q$ on the synchronization behaviors and synchronization transition of ring neuronal networks constructed by fractional-order neuronal models.

The firing mode with different parameter $\beta$ when $k_1 = 0.4, C = 10$ is plotted in Figure 16, and the other parameters are set as mentioned in Section 1. Figure 16 shows the firing mode of the first neuron when the parameter $\beta$ is different. From the numerical simulations, we can find that the neuronal models display chaotic bursting, spiking firing, and periodic bursting when $-0.06 < \beta < 0.02$, $\beta < -0.06$, and $\beta > 0.02$. In this section, the synchronization behaviors in the three regions of $\beta$ are studied numerically.

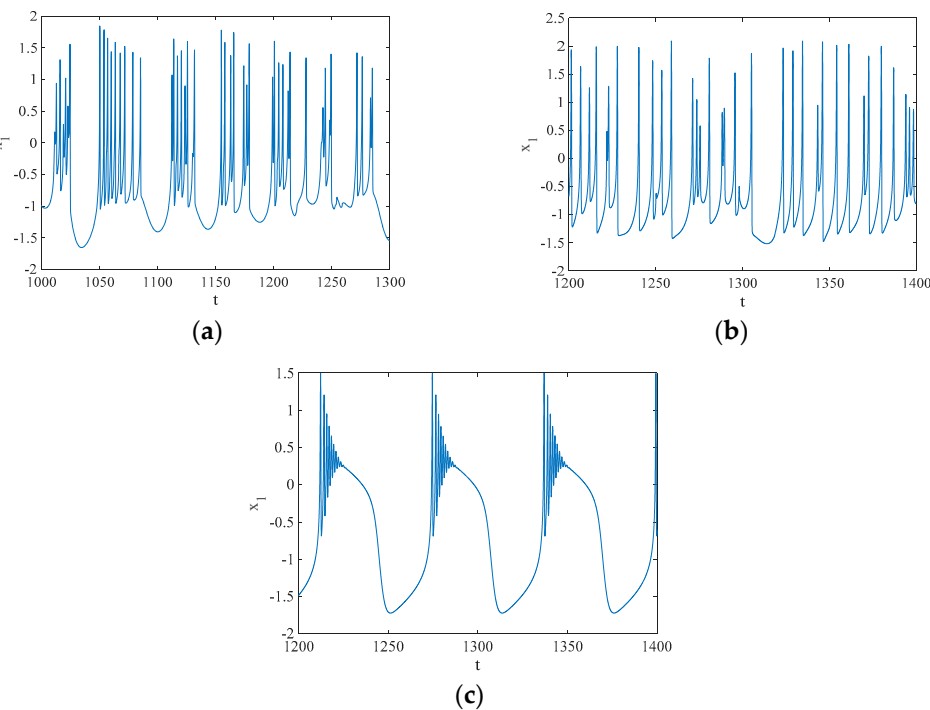

**Figure 16.** Corresponding time series of $x_1$ for $q = 0.8$ (**a**) $\beta = -0.02$, (**b**) $\beta = -0.08$, and (**c**) $\beta = 0.04$.

4.2.1. Synchronization Behavior of the Neuronal Network When $\beta > 0.02$

In this part, the parameter $\beta = 0.04$ is set as an example. The curves of $R \sim C$ when $q = 0.8$ for different $k_1$ are shown in Figure 17. From Figure 17, we find the effects of parameter $k_1$ and coupling strength on synchronization behaviors. The synchronization factor increases with increasing coupling strength and $k_1$. When $\beta = 0.04$, $k_1$ plays a dominant role in the network synchronization, because the network cannot reach perfect synchronization when $k_1 = 0.15, 0.2, 0.25, 0.3$ and the coupling strength ranges from 1 to 16. The state of the network can be transformed from asynchronization to perfect synchronization with a minor change in $k_1$ when $k_1 > 0.35$.

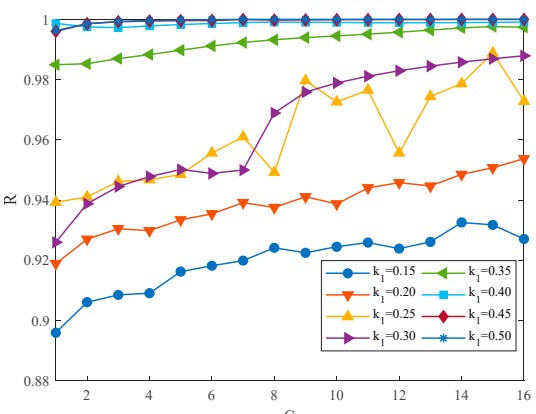

**Figure 17.** Curves of $R \sim C$ when $q = 0.8$ for different $k_1$.

To observe the effect of fractional order, $k_1$ is set as 0.35. Fractional order cannot influence the neuronal models' firing mode, but it can change the synchronization degree. Figure 18 shows the $R \sim q$ curves for different coupling strengths. We can conclude that the increase in fractional order weakens the synchronization of the neuronal network. The synchronization is robust against alterations of coupling strength when fractional order is small. When the fractional order is 0.7, the difference in the synchronization factor between

coupling strengths 1 and 5.5 is only 0.05, but when the fractional-order is 1, the difference becomes 0.2.

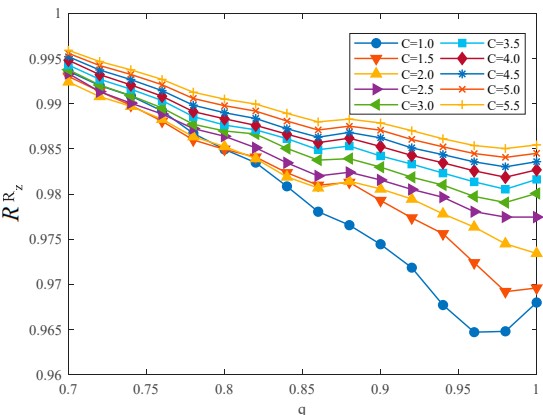

**Figure 18.** Curves of $R \sim q$ for different coupling strengths.

Figure 19 shows the neuronal network's spatiotemporal patterns and corresponding snapshots when $C = 4.5, k_1 = 0.4$. It is concluded that the strength of synchronization is higher when the fractional order is 0.7 than when the fractional order is 0.98. When $q = 0.7, C = 4.5$, the $x_{1i}$ are uniformly distributed values, so the network is in nearly perfect synchronization. When $q = 0.98, C = 4.5$, the $x_{1i}$ are disorderly. However, the slow variable's synchronization factor $R_z$ is calculated as 0.99, so the network is in bursting synchronization. The fractional-order can also induce the transition of perfect synchronization and bursting synchronization when the ring network is under electromagnetic radiation.

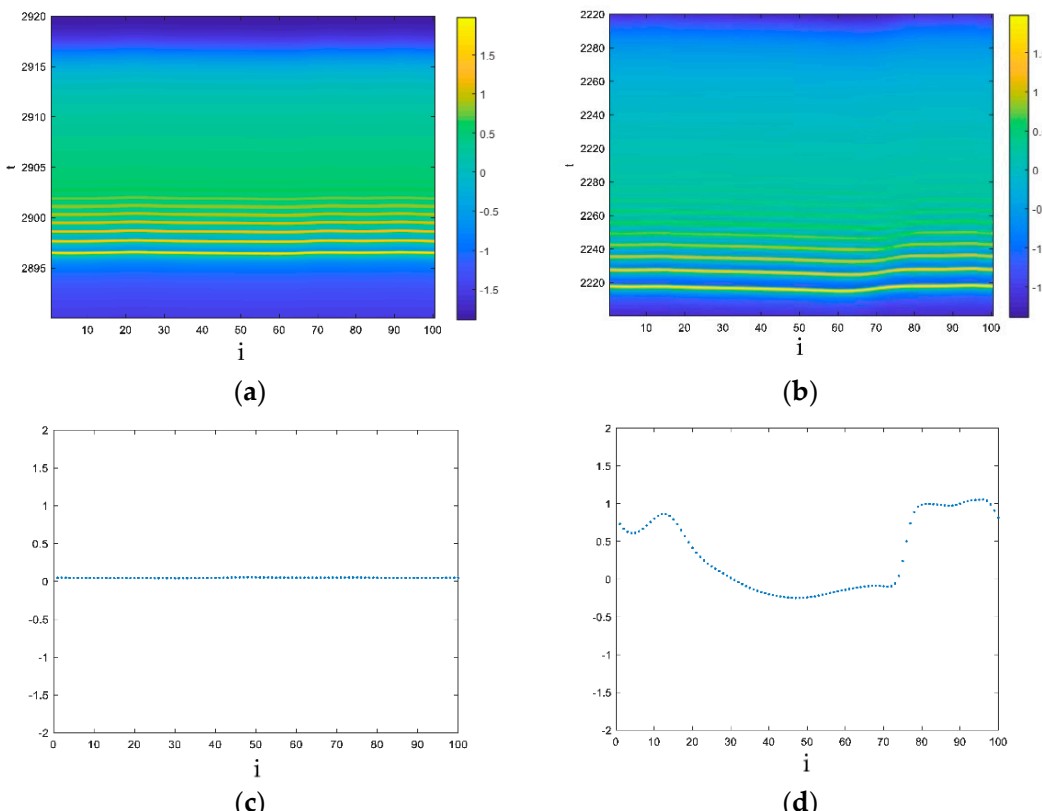

**Figure 19.** Neuronal network spatiotemporal patterns for (**a**) $q = 0.7$, and (**b**) $q = 0.98$. The corresponding snapshots for (**c**) $q = 0.7$, and (**d**) $q = 0.98$.

In addition, $k_1$ can also induce synchronization transition; that is, the firing mode of neuronal models changes for different $k_1$ when the network is in perfect synchronization. The parameters are set as $q = 0.8, C = 10$ and the neuronal network is in perfect synchronization. Figure 20 shows the phase diagram of $(z_i, x_i)$ when $k_1 = 0.4, 0.5, 0.6, 0.7$. As shown in Figure 20, a neuronal model in the network displays periodic-10 bursting, periodic-7 bursting, periodic-5 bursting, and periodic-4 bursting when $k_1 = 0.4, 0.5, 0.6, 0.7$. The number of spikes in one bursting decreases with increasing $k_1$, so the synchronization mode varies with $k_1$. The synchronization transition induced by $k_1$, which is perfect periodic-10 synchronization → perfect periodic-7 synchronization → perfect periodic-5 synchronization → perfect periodic-4 synchronization, is observed.

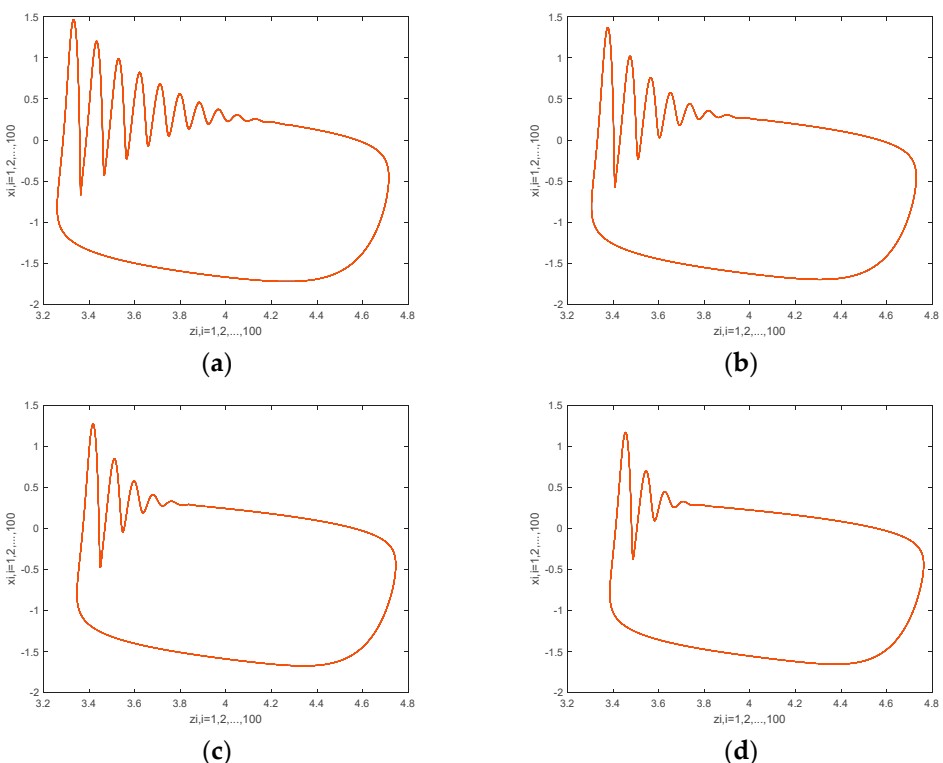

**Figure 20.** Phase diagram of $(z_i, x_i)$ for (**a**) $k_1 = 0.4$, (**b**) $k_1 = 0.5$, (**c**) $k_1 = 0.6$, and (**d**) $k_1 = 0.7$.

4.2.2. Synchronization Behavior of the Neuronal Network When $-0.06 < \beta < 0.02$

In this part, the parameter $\beta = -0.02$ is set as an example. The synchronization is different from the above analysis. As shown in Figure 21, the synchronization factor also increases with increasing coupling strength and $k_1$, but we can conclude that it is difficult to reach perfect synchronization. The coupling strength ranges from 1 to 16, and the synchronization factor just reaches approximately 0.3.

The neuronal network's spatiotemporal patterns and corresponding snapshots at $t = 2500s$ when $k_1 = 0.4$ are shown in Figure 22. It is found that the $x_{1i}$ are disorderly when the coupling strength is 1 and 16. The degree of synchronization is higher when the coupling strength is 16 than when the coupling strength is 1, but they are not synchronized. After calculating the slow variable's synchronization factor, the network is not in bursting synchronization. This part of the view is different from the case of the network without electromagnetic radiation, as shown in Figure 15d. Under the same fractional-order and coupling strength, the network can be in perfect synchronization.

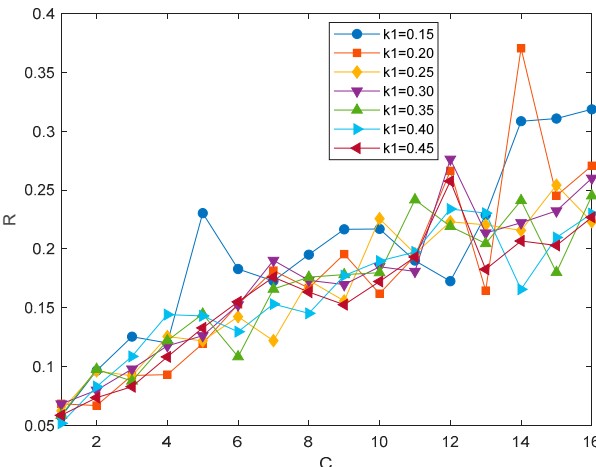

**Figure 21.** Curves of $R \sim C$ for different $k_1$.

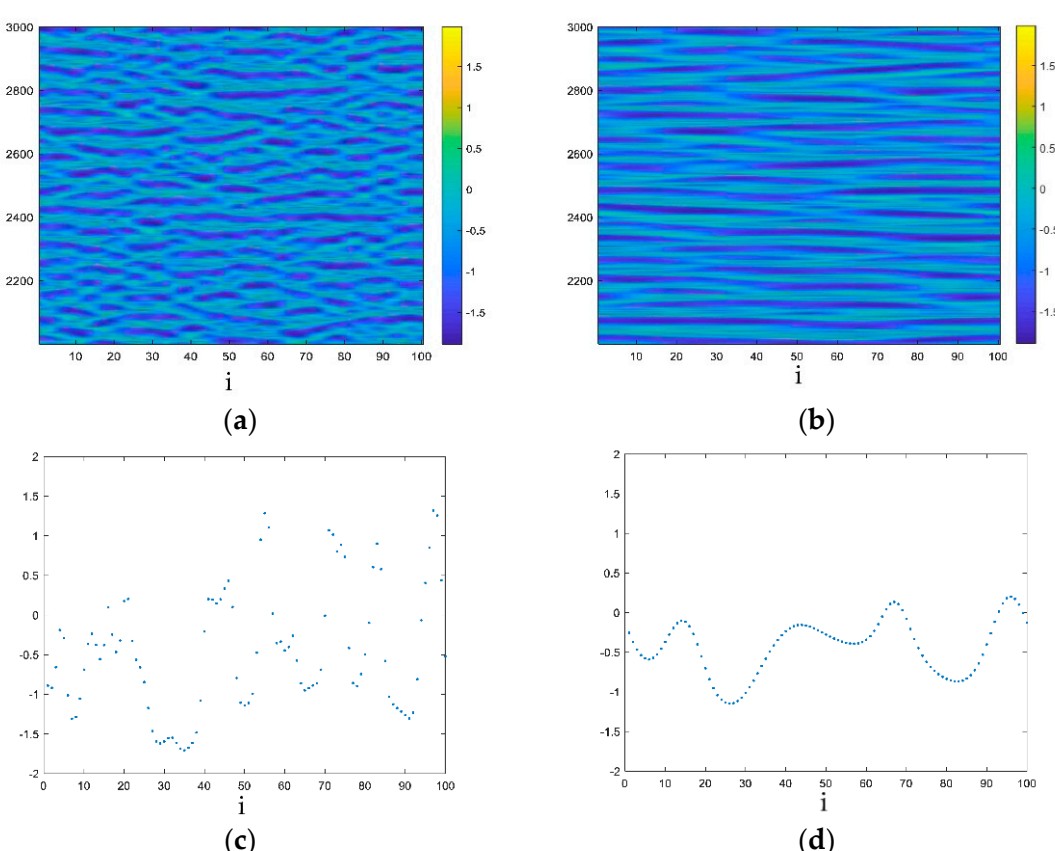

**Figure 22.** Neuronal network spatiotemporal patterns for $k_1 = 0.4$ (**a**) $C = 1$, and (**b**) $C = 16$. The corresponding snapshots at $t = 2500s$ for $k_1 = 0.4$ (**c**) $C = 1$, and (**d**) $C = 16$.

### 4.2.3. Synchronization Behavior of the Neuronal Network When $\beta < -0.06$

In this section, the parameter $\beta = -0.08$ is set as an example. Some novel phenomena are observed. The $R \sim C$ curves when $k_1 = 0.4$ for different fractional-orders are plotted in Figure 23. We find that the network is in perfect synchronization when $C > 7$, but the neuronal models' dynamic behaviors are different when $6.4 < C < 7$ for different fractional orders. In addition, the transition of synchronization and asynchronization is abrupt when the fractional-order and coupling strength change slightly. The synchronization factor changes from 0.1 to 1 directly. This phenomenon is not found in previous studies.

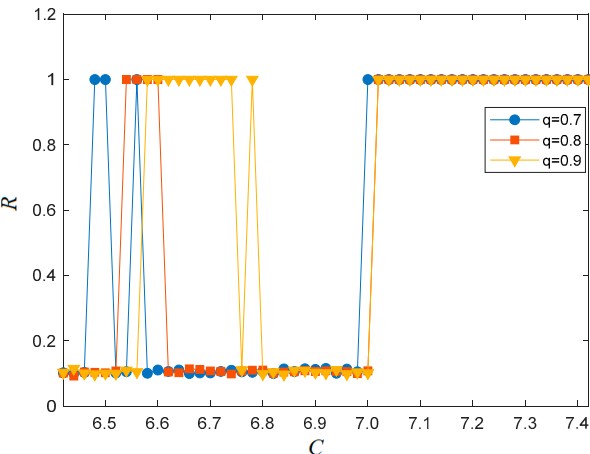

**Figure 23.** Curves of $R \sim C$ when $k_1 = 0.4$ for different $q$.

Figure 24 shows the neuronal network spatiotemporal patterns at different fractional orders and coupling strengths. As shown in Figure 24, for each fractional order, the network undergoes several sudden transitions of asynchronization and perfect synchronization with the variation in coupling strength. The dynamic behavior of neuronal models in the network is different at different fractional-orders and coupling strengths. When $C = 6.7$, the network is asynchronized at $q = 0.7$ and $q = 0.8$, but the network is perfectly synchronized at $q = 0.9$. When $C = 6.48$, the network is in perfect synchronization at $q = 0.7$, but the network is in asynchronization at $q = 0.9$ and $q = 0.8$. When $\beta < -0.06$, fractional-order can also induce a synchronization transition. We can find that the larger the fractional-order is, the larger the range of asynchronization, as shown in Figure 23. From the value of the slow variable's synchronization factor when the network is not in perfect synchronization, the network cannot obtain bursting synchronization. In the neuronal network without electromagnetic radiation, we could not find the abrupt changes in the synchronization factor and synchronization behaviors.

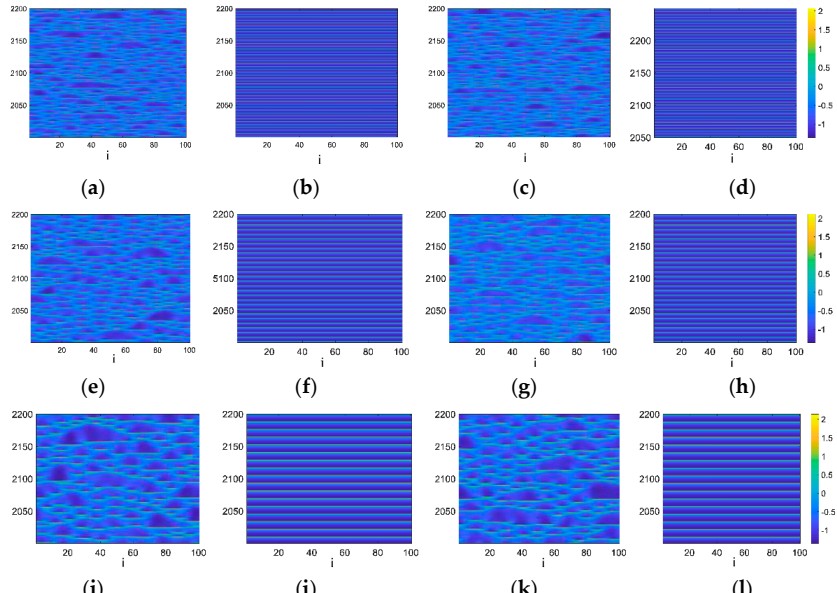

**Figure 24.** Neuronal network spatiotemporal patterns and corresponding snapshots for different fractional-orders and coupling strengths. (**a**) $q = 0.7, C = 6.43$, (**b**) $q = 0.7, C = 6.48$, (**c**) $q = 0.7, C = 6.7$, (**d**) $q = 0.7, C = 7.1$; (**e**) $q = 0.8, C = 6.48$, (**f**) $q = 0.8, C = 6.6$, (**g**) $q = 0.8, C = 6.7$, (**h**) $q = 0.8, C = 7.1$; (**i**) $q = 0.9, C = 6.48$, (**j**) $q = 0.9, C = 6.7$, (**k**) $q = 0.9, C = 6.9$, (**l**) $q = 0.9, C = 7.1$.

For the influence of $k_1$, as shown in Figure 25, the synchronization degree increases when $k_1$ increases, but the network has difficulty obtaining perfect synchronization. The parameters are set as $q = 0.8, C = 6.7$.

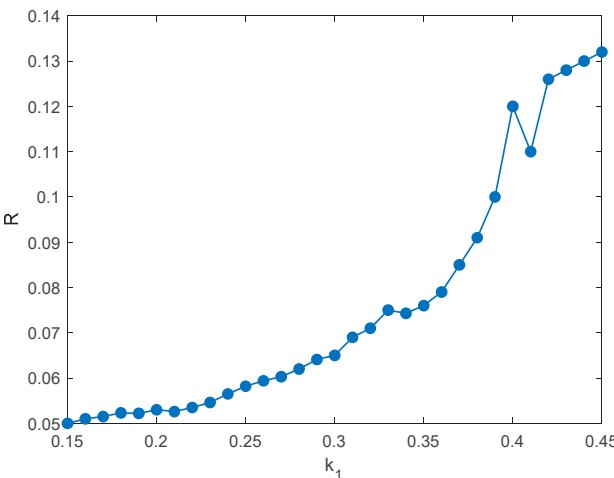

**Figure 25.** $R \sim k_1$ for $q = 0.8$ and $C = 6.7$.

## 5. Conclusions

This paper investigates the synchronization behaviors and synchronization transitions of fractional-order neuronal networks under electromagnetic radiation.

(1) For the two coupled neuronal models under electromagnetic radiation:

    (a) With increasing fractional-order, the synchronous threshold of the coupling strength fluctuates first, then increases and finally decreases (in [34] (without electromagnetic radiation), the threshold of coupling strength only increases first and then decreases with increasing fractional-order).

    (b) The synchronization transitions of the coupled fractional-order neuronal models, which contain bursting synchronization, perfect synchronization, and phase synchronization, are observed when the fractional-order or parameter $k_1$ changes.

    (c) In addition, when the two coupled neuronal models are in perfect synchronization, the transition of perfect chaotic synchronization and perfect periodic synchronization is observed when changing the fractional order or parameter $k_1$. From the ISI bifurcation diagram in Figure 12, when $q = 0.6$, the system has more diverse synchronization modes, which are perfect chaotic synchronization, perfect periodic-6 synchronization, perfect periodic-4 synchronization, perfect periodic-3 synchronization, and perfect periodic-2 synchronization when the value of $k_1$ is different. However, when $q = 0.8$, only perfect periodic-2 $\rightarrow$ spiking synchronization occurs with increasing $k_1$. Compared with [28], more diverse synchronization behaviors and synchronization transition induced by fractional order and other parameters were found, like the synchronization transition of phase synchronization, perfect synchronization and bursting synchronization, and our work shows more details of the synchronization behaviors of coupled fractional-order neuronal networks under electromagnetic radiation.

(2) For the ring network constructed by fractional-order HR models without electromagnetic radiation:
The fractional-order can also change the degree of synchronization, similar to the other influencing factors reported in [22–34], and the transition of periodic synchronization and chaotic synchronization induced by the fractional-order is observed.

(3) For the same ring network under electromagnetic radiation.

Different from the results of [37] in which only the parameter $k_1$ changes the firing activities of neuronal models in the network, this paper focuses on the influence of the parameters $\beta$, $k_1$ and fractional-order $q$ on the synchronization behaviors and synchronization transitions. Obviously different from the integer-order neuronal network and the fractional-order neuronal network without electromagnetic radiation, the fractional orders cannot change the firing activity of a single neuronal model in the fractional-order neuronal network with electromagnetic radiation. However, $q$ can influence the synchronization degree of ring fractional-order neuronal networks.

(a)   When $\beta > 0.02$, the synchronization degree decreases with increasing fractional-order. The parameter $k_1$ can induce the synchronization transition of perfect periodic-10 synchronization, perfect periodic-7 synchronization, perfect periodic-5 synchronization, and perfect periodic-4 synchronization.

(b)   When $-0.06 < \beta < 0.02$, it is difficult for the network to reach synchronization, and the fractional order has difficulty changing the synchronization degree.

(c)   In particular, when $\beta < -0.06$, the network has a sudden transition of asynchronization and perfect synchronization. The synchronization factor goes suddenly from 0.1 to 1. The larger the fractional order is, the larger the range of asynchronization is. The synchronization degree increases with increasing $k_1$.

**Author Contributions:** Conceptualization, X.Y. and G.Z.; methodology, X.Y.; validation, X.Y.; data curation, X.Y.; writing—original draft preparation, X.Y.; writing—review and editing, G.Z. and D.W.; visualization, X.Y.; supervision, X.L. All authors have read and agreed to the published version of the manuscript.

**Funding:** This research received no external funding.

**Institutional Review Board Statement:** Not applicable.

**Informed Consent Statement:** Not applicable.

**Data Availability Statement:** The data used to support the findings of this study are included within the article, and other data used can be obtained from the corresponding author upon request.

**Conflicts of Interest:** The authors declare no conflict of interest.

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
