# Peer review of "The Synchronization Behaviors of Coupled Fractional-Order Neuronal Networks under Electromagnetic Radiation"

_symmetry, doi:10.3390/sym13112204_

Round 1

Reviewer 1 Report

The manuscript under review is devoted to the study of the issue of synchronization of two physical neural networks by modeling the process with a certain dynamic system. The mathematical formulation of the problem is a system of fractional differential equations. The solution of this system is investigated. The main task is to synchronize two networks exposed to electromagnetic radiation. The results of the study are based on the statements proved in the work and the studies of the numerical solution carried out. All results are illustrated with diagrams that are consistent with the findings. I think the manuscript should be published.

Author Response

Thank you for your comments.

Reviewer 2 Report

The work "The synchronization behaviors of coupled fractional-order neuronal networks under electromagnetic radiation"is an extensive study on  Neuronal networks synchronization under electromagnetic radiation. The work is sound and well structured but there are few comments that the authors should address before to accept this article.

1) It would be very helpful include in the introduction a discussion about some experimental results correlated with the theoretical results obtained from all the models presented in the section. I know the authors mentioned this correlation but briefly.

2) Are the theoretical results presented here have an experimental counterpart?  Are there any experimental evidence of the effect of electromagnetic radiation on these type of synchronization behaviors?

3) Since this work is about electromagnetic radiation, the introduction of this concept is not clear. What kind of electromagnetic field is this? Again, all these results seems not connected with real experimental conditions.

Could you please introduce and or clarify these concerns on the draft.

Minor issue: There are some sentences whose style make it difficult to understand. For instance, page 2, lines 67-69. Or, page 2, lines 88-89. Or, the typo in page 13, lines 365.

Reviewer 3 Report

The paper presents a theoretical study of coupled fractional order neural networks. The paper is well written but the authors should once again read the text and remove minor spelling mistakes scattered through the paper. Otherwise, I support publishing the paper.

1. What is the main question addressed by the research?
The subject of research is a study of the synchronization behaviors of coupled fractional-order neuronal networks under electromagnetic radiation. The authors performed theoretical study of the behaviour of the system under various sets of parameters with special focus on synchronisation.

2. Do you consider the topic original or relevant to the field? Does it
address a specific gap in the field?
To my best knowledge the topic and study is original and thus fill a gap in the field.

3. What does it add to the subject area compared with other published
material?
The authors should clarify the difference between the current contribution and the position 26 in the references list.

4. What specific improvements should the authors consider regarding the
methodology? What further controls should be considered?
None in my opinion.

5. Are the conclusions consistent with the evidence and arguments
presented and do they address the main question posed?
yes

6. Are the references appropriate?
yes

7. Please include any additional comments on the tables and figures.
In general the paper is well written but some typos and grammar mistakes are left in the manuscript. For instance:
1. sentence starting in line 11
2. sentence starting in line 95
3. sentence starting in line 150
etc

Also as someone not specializing in this field I cannot see the ''electromagnetic radiation'' element in the research.
I was expecting an experimental element in research but it is not present. Could the authors explain how
electromagnetic radiation is included in the considered model? 

Reviewer 4 Report

Dear Authors,
i propose the following changes to the article:
- add a description of each section of the article in the introduction,
- a section should be added to the article: literature review,
- the article should conclude with bulleted conclusions,
- please consider referencing the Hodgkin-Huxley model.
Good luck

Round 2

Reviewer 4 Report

Dear Authors,

I have carefully read the changes made to the article and I believe that:

- further, no literature review was added as a separate section of the article,

- there are no bulleted conclusions from the research in the summary.

Greetings

Author Response

We are so sorry that our modification did not meet your requirements. We now divide the introduction into two parts: Literature review and Description of each section, and the conclusion includes three parts. Please see the new version.